# CoLLAT: On Adding Fine-grained Audio Understanding to Language Models using Token-Level Locked-Language Tuning

**Amila Silva**[1,2]     **Spencer Whitehead**[1]     **Chris Lengerich**[1]     **Hugh Leather**[1]

[1]Meta AI (FAIR)
[2]The University of Melbourne
amilas@unimelb.edu.au, {srw5,clengerich,hleather}@meta.com

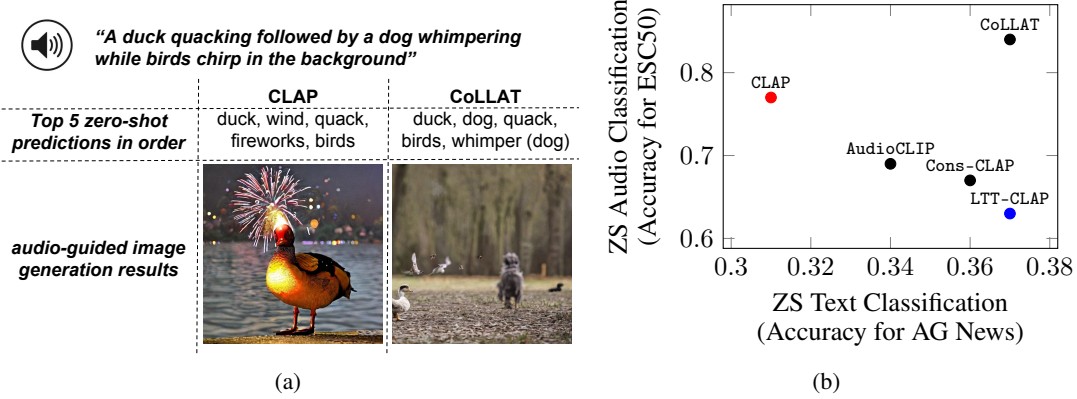

(a)                                             (b)

Figure 1: CoLLAT yields fine-grained audio-to-text grounding by token-level alignment of audio-text with a locked pretrained text embedding space, which (a) enables fine-grained audio understanding and novel multimodal capabilities (e.g., audio-guided image generation); and (b) achieves superior performance across both unimodal (audio, text) and multimodal (e.g., audio and text) settings.

## Abstract

Humans can easily understand various audio concepts, but conventional audio classification models fail due to their inability to predict unseen classes during training. To address this challenge, recent literature has explored contrastive language-audio pretraining to learn an audio understanding model using natural language supervision from a pretrained language model. However, despite their reasonable zero-shot performance in audio understanding, these models typically fail to achieve optimal performance while preserving the text understanding capabilities of the pretrained language model. They also perform poorly when comprehending audio clips with multiple audio concepts. To bridge these gaps, we propose CoLLAT: **Co**ntrastive **L**ocked **L**anguage and **A**udio **T**uning. This is a framework to effectively learn an audio understanding model with a locked language model, which is learned using a novel pretraining objective for audio-to-text grounding to yield fine-grained audio understanding. Our extensive experiments, which include several downstream applications such as audio classification, cross-modal retrieval, audio captioning and audio-guided image generation, demonstrate that CoLLAT yields state-of-the-art performance for audio understanding. Additionally, it unlocks audio guidance to applications built on top of pretrained language models.

37th Conference on Neural Information Processing Systems (NeurIPS 2023).

# 1 Introduction

**Motivation.** The sound perception system in humans is capable of understanding complex audio concepts and interpreting them in a way that allows us to interact with our environment effectively [21]. For example, when humans listen to a sound clip of a duck quacking followed by a dog whimpering while birds chirp in the background (see example in Fig. 1a), they can easily discern various audio concepts (e.g., a duck quacking, a dog whimpering, birds chirp) in that sound clip, enabling them to construct a holistic understanding of the sound clip. However, conventional audio classification models [22, 24, 30, 10, 25] that aim to associate audio recordings with one or more categories from a set of predefined categories fail to be competitive with the human auditory system due to their inability to predict unseen classes during training. To address this issue, recent literature has explored contrastive language-audio pretraining [13, 7] to learn an audio understanding model using natural language supervision. These methods aim to learn an audio encoder and a text encoder simultaneously, which produce a joint embedding space for audio and text. The joint embedding space is learned to preserve the correspondence of audio-text pairs present in the pretraining dataset. Such a joint embedding space creates an open vocabulary for audio-language correspondence, enabling new audio-language capabilities such as zero-shot audio classification and audio+language guided image generation. However, existing works on contrastive language-audio pretraining [13, 35, 7] lack the following strengths.

First, previous works [13, 35, 7] typically initialize their text encoder using CLIP [27] – a strong pre-trained text encoder. However, these techniques [7, 13] often fail to achieve optimal performance for audio-language understanding without fine-tuning the text encoder (i.e., by keeping the text encoder frozen). In Fig. 1b, we report the performance of a set of baselines, where CLAP, AudioCLIP, and Cons-CLAP models fine-tune the pre-trained CLIP text encoder, while the rest keep the text encoder frozen. As can be seen, fine-tuning the text encoder improves cross-modal (e.g., zero-shot audio classification) performance, but it comes at the cost of losing its language understanding capabilities. This could be attributed to the lack of sufficient textual information in publicly available audio-text datasets (e.g., AudioSet [8]), which are relatively smaller in size compared to the datasets used to originally pretrain CLIP[1]. In this work, ***we study the possibility of achieving SOTA performance for audio-text understanding with a pretrained text encoder that remains locked.*** Incorporating audio understanding without tuning the CLIP text encoder is not only beneficial for capitalizing the pre-trained text encoder's existing capabilities, but it also enables audio or audio+text guidance to any AI application that uses text-guidance from the CLIP text encoder (e.g., CLIP-guided image generation) without needing to retrain the application-specific model (e.g., diffusion model of a CLIP-guided image generation pipeline). Recent studies on language-visual pretraining [3, 4] have demonstrated that matching the sizes of the encoders is vital for achieving strong cross-modal performance while keeping one encoder locked. Nevertheless, most existing audio-text pretraining methods use audio encoders that are substantially smaller in size compared to the text encoder. Building upon this observation, we propose an audio-text pretraining architecture that jointly scales both the audio and language encoders, allowing for strong cross-modal performance with locked language tuning.

Second, existing contrastive audio-text pretraining techniques [35, 7, 13] have limited efficacy when understanding complex audio clips with multiple audio concepts (see results in Table 2). This shortcoming could be attributed to the absence of explicit fine-grained cross-modal grounding in these techniques, as they solely depend on global embeddings – i.e., a single vector summarizing the semantic content of a given text/audio input, to maintain the correspondence between audio and text. Although such global embedding alignment methods achieve reasonable performance in audio understanding, encoding audio and text into global embeddings will lose much fine-grained information, which is critical to distinguish hard audio-text pairs. In contrast, the token-level embeddings of the intermediate layers of transformer-based encoders for text and audio possess better understanding of fine-grained concepts as they have one or more token-level embeddings dedicated to each fine-grained concept in an audio/text input. Despite the knowledge in such token-level embeddings are explicitly exploited in other domains for fine-grained cross-modal grounding (e.g., text-image [34, 37] and text-video [39]), to the best of our knowledge, none of the previous audio-text pretraining techniques have explored this. To bridge this research gap, ***this work studies the importance of exploiting token-level audio-text alignment along with the global-level alignment to yield better fine-grained understanding of audio-text correspondence.***

---

[1]CLIP's original pretraining dataset is almost 200 times larger than AudioSet.

**Contributions.** The contributions of this work are to propose:

- A neural architecture for effective contrastive audio-language pretraining that yields SOTA performance for audio-text understanding while keeping the text encoder locked.
- An improved audio-text pretraining objective function that explicitly encourages the model to learn fine-grained audio-text grounding to unlock complex audio comprehension.

We verify the superiority of our proposed framework using a diverse set of experiments – zero-shot audio classification, cross-modal retrieval, audio captioning and audio-guided image generation, which shows that our framework yields SOTA performance for both unimodal (audio or text) and multimodal (audio and text) understanding, while capitalising on the existing capabilities of the text encoder (e.g., text-guided image generation) and alleviating its substantial cost of re-training.

## 2 Related Work

### 2.1 Audio-to-Text Grounding

Audio-to-text grounding refers to establishing a connection between textual concepts and audio concepts. In earliest research efforts [22, 24, 30, 10, 25], the audio-to-text grounding problem was approached as a classification task that utilized machine learning models to link audio recordings to pre-defined class labels. These approaches explored various machine learning models, spanning from traditional machine learning models such as Support Vector Machine [19, 31, 32], to advanced neural architectures like Convolutional Neural Networks (CNN) [26, 23, 14] and transformers [10, 18]. These models were designed to operate on either static [22, 24, 30] or trainable [12] time-frequency transformations of raw audio. While these approaches were successful in predicting the class labels used during training, they often struggled to predict novel audio concepts.

To address this challenge, recent works [35, 13, 7, 5] have explored that how to learn a joint embedding space to preserve the correspondence between audio and text with the language supervision from pretrained text encoders. This approach enables an open vocabulary for audio concepts to unlock applications such as zero-shot audio classification. These works typically employ two separate encoders: one for audio and one for text. Each encoder maps the given input, such as an audio clip or text prompt, to an embedding vector that preserves the knowledge of the input. These models learn the parameters of the encoders using a contrastive loss function [27], which aims to ensure that the embeddings of corresponding text-audio pairs are similar to each other while pushing the embeddings of dissimilar pairs further apart.

To exploit the language comprehension abilities of large pretrained text encoders [27, 28], these studies often initialize the text encoder with a pretrained model like CLIP [27]. These text encoders typically have billions of parameters and are trained using large-scale datasets such as LAION. For the audio encoder, different works adopt various architectures – Wav2CLIP adopts ResNet-based architecture [1], AudioCLIP employs ESResNeXt [12], and CLAP uses CNN14 [17]. Then, these works jointly train the audio and text encoder to preserve audio-text correspndence. In addition to the differences in audio encoder architecture, these works vary in how they construct text prompts and their use of auxiliary objectives. For example, AudioCLIP [13] creates text prompts by concatenating the discrete class labels of an audio clip from datasets like AudioSet [9]. On the other hand, CLAP [7] utilizes a set of audio-text datasets that provide semantically and syntactically meaningful captions for each audio clip to train their model. Additionally, AudioCLIP [13] incorporates auxiliary objectives to preserve audio-image and image-text correspondences along with the audio-text correspondance.

Despite these differences, all these works fail to yield optimal performance for audio-text understanding without fine-tuning the text encoder – i.e., keeping the weights of the pretrained text encoder. Previous studies [4, 3] suggest that this may be due to the size mismatch of the audio and text encoders observed in previous works [7, 13] (see Figure 2a). As shown in Fig. 2b, simply scaling the audio encoder to match the size of the text encoder does not result in optimal performance too, especially given the relatively small dataset available with audio-text pairs. To address this issue, recent studies [3, 4] in other multimodal domains propose the use of neural architectures [4], that enable joint scaling of the sizes of multimodal encoders. Nevertheless, to the best of our knowledge, such architectures have not been explored in the context of audio-text understanding.

To bridge this gap, our work proposes a novel neural architecture for audio-text pretraining that solves the size mismatch problem in existing works, enabling our model to achieve optimal performance for

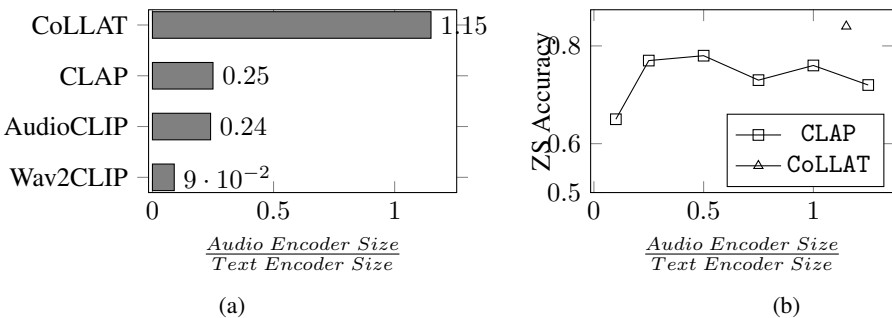

(a)                                      (b)

Figure 2: (a) Size ratio of audio and text encoders across different methods; and (b) zero-shot audio classification performance of CLAP (using FSD50K) with different audio encoder sizes, while keeping the size of the text encoder fixed to that of CoLLAT.

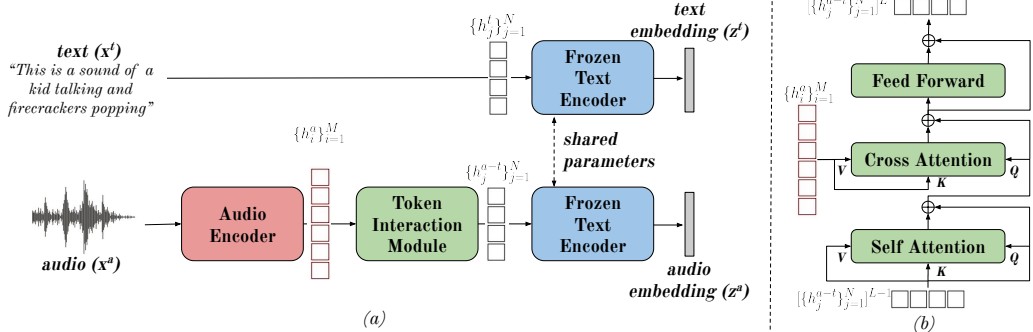

Figure 3: (a) Overview of `CoLLAT`, consisting of the Token Interaction Module for mapping audio tokens to text tokens by adopting a series of blocks as detailed in (b).

audio-to-text understanding without the need for fine-tuning the text encoder. The strenghts of our architecture is three-fold: (1) it preserves the language understanding capabilities of the pretrained text encoder, resulting in better audio-to-text grounding even with a relatively small audio-text dataset; (2) it is compatible with text encoders of any scale, making it more viable for learning audio understanding using evolving language models; and (3) it maps audio embeddings into the same representation space as the pretrained text encoder, introducing audio-guided controllability to any downstream application that relies on text guidance from the pretrained text encoder.

## 2.2 Token-level Grounding

Existing audio-text contrastive pretraining techniques [35, 7, 13] do not explicitly consider the fine-grained alignment of audio concepts and textual concepts. These methods are only trained to align the global embeddings (i.e., last layer output from the encoders which summarizes all the concepts in the corresponding input) from the encoders. Such global embeddings may not be able to capture the individual concepts in complex audio clips containing multiple audio concepts. In contrast, the token-level hidden representations of audio/text encoders can effectively represent fine-grained concepts. Therefore, it is crucial to have token-level alignment between audio and text to achieve a more fine-grained understanding of audio.

This research challenge has been deeply studied in text-visual domain. For example, the works in [39, 2, 34, 37] have shown the importance of aligning the token-level hidden representation of the visual and text encoders to yield fine-grained visual understanding. The same concept has also been shown to be important for similar other learning tasks such as knowledge distilliation [38, 33]. However, token-level contrast between audio and text modalities has not been well-studied in previous audio-text pretraining methods. Some works [16, 11] have focused on explicit token-level attention for audio-related downstream applications like audio captioning, but they are limited to specific tasks. Therefore, our work introduces token-level alignment as an auxiliary task for conventional contrastive audio-text pretraining to yield fine-grained audio understanding.

# 3 CoLLAT

Our model aims to learn fine-grained audio understanding with the help of a pretrained text encoder, without fine-tuning the text encoder. Unlike existing techniques that use separate audio and text encoders, our model shares the pretrained text encoder, which remains frozen during training, for both audio and text encoding. The main strengths of this architecture are twofold: (1) it allows us to use large pretrained language encoders (with billions of parameters) for the text encoder without introducing a significant mismatch between the encoders used for audio and text, as this architecture scales the encoder size for each modality jointly; and (2) it enables us to leverage the implicit knowledge (i.e., intermediate layers) in the text encoder to encode audio. By using a frozen shared text encoder for both audio and text encoding, the corresponding audio-text pairs are forced to implicitly align their token-level embeddings to produce similar global embeddings. Figure 3 (a) shows a high-level schematic of the proposed model architecture.

We reuse the pretrained CLIP text encoder [27] as our text encoder, but our model is compatible with any other pretrained text encoder at any scale. This text encoder expects the static token embeddings of the given text prompt as input. To encode audio in a way that is compatible with the pretrained text encoder, we adopt a transformer-based audio encoder [10] to produce token-level embeddings for the audio tokens (i.e., patches in the log mel spectrogram) and a cross-attention-based token interaction module to produce corresponding text token embeddings from the audio token embeddings. As shown in Fig. 3 (a), our model takes either audio and/or text as inputs and produces corresponding embeddings while preserving audio-text correspondence. We formally define our architecture and training objectives below.

Let $\mathbb{D} = \{X^a, X^t\}$ be a set of audio-text pairs, where each pair is represented as $\{x^a, x^t\}$. Here, $x^a \in \mathbb{R}^{F \times T}$ is the processed audio clip represented as a log mel spectrogram with $F$ spectral components (e.g. Mel bins) and $T$ time bins; and $x^t$ is the tokenized text prompt explaining the audio concepts in $x_i^a$.

## 3.1 Model Architecture

For a given $\{x^a, x^t\}$, we preprocess $x^a$ as a sequences of patches of its 2D audio spectrogram and $x^t$ as a sequence of tokens by tokenizing the text prompt via the pretrained BPE-based tokenizer in CLIP. We denote these preprocessing functions for audio and text as $preprocess_a()$ and $preprocess_t()$, respectively, and the output from these functions as $\{x_i^a\}_{i=1}^M$ and $\{x_j^t\}_{j=1}^N$, respectively.

$$\{x_i^a\}_{i=1}^M = preprocess_a(x^a), \ \{x_j^t\}_{j=1}^N = preprocess_t(x^t)$$

Here, $M$ and $N$ denote the maximum lengths of the audio and text sequences, respectively. The audio tokens $\{x_i^a\}_{i=1}^M$ are embedded using a transformer-based audio encoder [10] $f_a()$, which returns latent representations of the audio tokens $\{h_i^a\}_{i=1}^M$. The text tokens $\{x_j^t\}_{j=1}^N$ are initially embedded using the static embedding matrix $E_t$ of the pretrained text encoder.

$$\{h_i^a\}_{i=1}^M = f_a(x^a), \ \{h_j^t\}_{j=1}^N = E_t \cdot x^t$$

The audio token embeddings are then passed through an audio-text token interaction module $f_{a-t}()$, which aims to predict the corresponding text token embeddings $\{h_j^t\}_{j=1}^N$ using the audio token embeddings $\{h_i^a\}_{i=1}^M$ through fine-grained token alignment. The output from this module is denoted as $\{h_j^{a-t}\}_{j=1}^N$ since it has the same number of tokens as the text token embeddings.

$$\{h_j^{a-t}\}_{j=1}^N = f_{a-t}(\{h_i^a\}_{i=1}^M)$$

Our token interaction module consists of multiple cross-attention-based blocks [37] (see Fig.3 (b)) in series. These blocks take the currently predicted text token embeddings from the previous block as input and attempt to denoise them using the audio token embeddings of the corresponding audio clip. For the first block in the interaction module, we pass a set of random vectors as input, which are randomly initialized following a normal distribution with a mean of zero and a standard deviation of 1. This series of token interaction blocks is analogous to a denoising pipeline[15] that is guided by the audio token embeddings to produce the corresponding text token embeddings.

Finally, the global audio and text embeddings are produced by encoding $\{h_j^t\}_{j=1}^N$ and $\{h_j^{a-t}\}_{j=1}^N$ using the pretrained text encoder $f_t()$.

$$z^a = f_t(\{h_j^{a-t}\}_{j=1}^N), \ z^t = f_t(\{h_j^t\}_{j=1}^N)$$

Our framework is trained using the following objective functions:

## 3.2 Cross-modal Token-Level Alignment Loss

To achieve the fine-grained grounding between audio and text while making CoLLAT agnostic to the order of the classes presented in the text, this loss function aims to find one-to-one mapping between text token embeddings $\{h_j^t\}_{j=1}^N$ from text and the produced text token embeddings $\{h_j^{a-t}\}_{j=1}^N$ using audio and make those token-level embeddings close to each other. We formulate the cross-modal token-level loss function as follows:

$$L_{cross-token} = ||g(\{h_j^{a-t}\}_{j=1}^N) - \{h_j^t\}_{j=1}^N||_1 \tag{1}$$

where $g$ denotes the function that reorder the token-level embeddings of an instance $\{h_j^{a-t}\}_{j=1}^N$ such that the cross-modal token-level loss function is minimized, which is pre-computed before each weight update. Given the time complexity of finding the optimal solution to $g$ in Eq. 1, we greedily compute $g$ as follows: given the token embeddings of $\{h_j^t\}_{j=1}^N$ and $\{h_j^{a-t}\}_{j=1}^N$, we initiate the process by starting from the right-most token embedding in $\{h_j^t\}_{j=1}^N$. This token is then paired with the closest token embedding in $\{h_j^{a-t}\}_{j=1}^N$. Next, we proceed to the second token embedding from the right in $\{h_j^t\}_{j=1}^N$, excluding the already mapped token embedding in $\{h_j^{a-t}\}_{j=1}^N$ from the possible set to be paired with the selected text token embedding. We iterate this process until we obtain the greedy one-to-one mapping between all the tokens in $\{h_j^t\}_{j=1}^N$ and $\{h_j^{a-t}\}_{j=1}^N$. We observed that this greedy approach could produce optimal re-ordering unless there are very similar classes in the same audio clip. Consequently, we do not anticipate a significant variation in the performance if the optimal reordering can be achieved, which we leave as future work given the time complexity of training the model with the optimal reordering.

## 3.3 Cross-modal Global Contrastive Loss

Motivated by [27], this objective function is introduced to preserve the global correspondence of the audio-text pairs in the training dataset. Given the global embeddings of a batch $B$ of audio and text pairs $\{Z_B^a, Z_B^t\}$, we compute the similarity metric between audio and text pairs in $B$ as:

$$S_B^{a-t} = \tau * (Z_B^a \cdot Z_B^t{}^T), \ \ S_B^{t-a} = \tau * (Z_B^t \cdot Z_B^a{}^T)$$

where $\tau$ is a temperature parameter to scale the range of logits. The similarity matrix $S \in \mathbb{R}^{|B| \times |B|}$ has $|B|$ correct pairs in the diagonal and $|B|^2 - |B|$ incorrect pairs in the off-diagonal, where $|B|$ refers to the batch size. We formulate our cross-modal global contrastive loss for the batch $B$ as follows:

$$L_{cross-global} = \frac{1}{2|B|} \sum_{i=0}^{|B|} \log[softmax(S_B^{a-t})_{(i,i)}] + \log[softmax(S_B^{t-a})_{(i,i)}] \tag{2}$$

## 3.4 Unimodal Token-Level Alignment Loss

Despite the reasonable performance of CoLLAT with the aforementioned two objective functions, we observed that the learned representations are not robust against weak perturbations of audio such as jitter (see ablation study in Section 5 for more details). To address this challenge, previous works [6, 40] adopts self-supervised objectives that aim to maximize the similarity of the embeddings of two different yet correlated views produced by perturbing the same audio input. Motivated by these works, we formulate our unimodal token-level loss for a given audio-text pair $\{x^a, x^t\}$ to improve the robustness our framework as follows:

$$L_{uni-token} = ||\{h_j^{a-t}\}_{j=1}^N - \{\tilde{h}_j^{a-t}\}_{j=1}^N||_1 \tag{3}$$

where $\{\tilde{h}_j^{a-t}\}_{j=1}^N$ is the token-level embeddings of the perturbed version of $x^a$ by adding a Gaussian noise. This loss forces the model to understand all the token-level embeddings (i.e., fine-grained concepts) of $x^a$ using its perturbed version.

**Overall Objective.** The final objective function of CoLLAT is as follows:

$$L = \lambda_1 \cdot L_{cross-token} + \lambda_2 \cdot L_{cross-global} + \lambda_3 \cdot L_{uni-token} \tag{4}$$

where $\lambda_1$, $\lambda_2$ and $\lambda_3$ control the importance of each loss term.

Table 1: Dataset Statistics

| Task | Audio Classification | | | | | Cross-modal Retrieval | | Audio Captioning | Audio-guided Image Gen. |
|---|---|---|---|---|---|---|---|---|---|
| Dataset | ESC-50 | UrbanSound8K | TUT | FSD50K | AudioSet | VGGSound | AudioCaps | AudioCaps | AudioCaps |
| # instances | 2K | 8K | 6.3K | 51K | 20.4K | 15K | 46K | 46K | 46K |
| # classes | 50 | 10 | 15 | 50 | 527 | 309 | N/A | N/A | N/A |
| Metric | Acc. | Acc. | Acc. | MAP | MAP | MRR | MAP@10 | COCO Metrics | N/A |

## 4 Experimental Setup

### 4.1 Datasets

**Training.** We adopt the AudioSet dataset as the training dataset, which consists of 2,041,792 audio and text label pairs collected using 10s long YouTube videos. Each audio clip can have multiple text labels to represent different audio concepts in different granular levels.

**Evaluation.** We evaluate our model using four main downstream applications: (1) audio classification; (2) cross-modal retrieval; (3) audio captioning; and (4) audio-guided image generation. We adopt 6 widely used real-world datasets to evaluate our model, which are shown in Table 1.

**Pre-processing.** We preprocess audio clips by representing them as 128-dimensional log Mel filterbank (fbank) features, computed with a 25ms Hamming window every 10ms. This results in a $128{\times}100t$ spectrogram as input to our audio encoder, where $t$ is the length of the audio clip in seconds. Next, we split the spectrogram into a sequence of N $16 \times 16$ patches with an overlap of 6 in both time and frequency dimensions, where $N$ is the number of patches and the effective input sequence length for the Transformer. We flatten each 16×16 patch to a 1D patch embedding of size 768 using a linear projection layer, which we refer to as the patch embedding layer. Since the Transformer architecture does not capture the input order information, and the patch sequence is also not in temporal order, we add a trainable positional embedding (also of size 768) to each patch embedding to allow the model to capture the spatial structure of the 2D audio spectrogram.

Following the findings in [7], we adopt the template of *"This is a sound of [class label 1], [class label 2], .... and [class label C]"* to generate a text prompt for an audio clip with $C$ number of class labels. Such templates have been shown to be more effective than simple concatenation [7]. Our decision to not utilize natural language descriptions or a keyword-to-caption augmentation [36] in favor of simple templates is due to the following reasons: (1) the cross-modal token-level loss function in CoLLAT aims to explicitly map each token in the given text prompt with its corresponding counterpart in the audio tokens (i.e., the patches in the Mel-Spectrogram of the audio). It may not be feasible to find such a mapping for certain tokens in natural language descriptions (e.g., stop words, adjectives) with the audio tokens. Consequently, having complex text prompts could adversely impact the training of CoLLAT; and (2) CoLLAT maintains the text encoder frozen during training, as it was pre-trained using text labels from the LAION dataset rather than natural language descriptions. Thus, attempting to produce text embeddings for natural language descriptions using the CLIP text encoder to train CoLLAT could introduce a data shift problem to negatively impact the training of CoLLAT.

**Training Details.** After performing a grid search, we set $\{\lambda_1, \lambda_2, \lambda_3\}$ as $\{1, 0.1, 0.01\}$. We utilize a GPU cluster comprised of 8 V100 GPU cards for training CoLLAT. With this hardware configuration, it takes approximately 320 GPU hours to train the model on AudioSet using the optimal hyperparameter setting.

## 5 Results

### 5.1 Audio Classification

Table 2 presents a comparison of CoLLAT with five different baselines in audio classification. The first baseline, called Supervise, trains a transformer from scratch without relying on features from a pretrained joint embedding space, while the other four baselines learn a joint embedding space between audio and text via contrastive pretraining. To conduct our experiments, we used five widely used datasets: ESC-50, US8K, TUT, FSD50K, and AudioSet, under two settings: zero-shot (ZS) and linear probe (LP). For zero-shot (ZS) setting, we first extract the embeddings for the audio clips and the possible set of labels using the audio and text encoder in each baseline. Then, we compute cosine distance between these text and audio embeddings to rank different class labels for a given audio clip.

Table 2: Results for Audio Classification. Here, ZS and LP denote zero-shot and linear-probe experimental setups for audio classification. For uni-label datasets – ESC-50, US8K and TUT, Accuracy is reported as the metric. multi-label datasets (FSD50K and AudioSet) adopts MAP (Mean Average Precision). Higher the value is better for both metrics.

| | ESC-50 (Acc) | | US8K (Acc) | | TUT (Acc) | | FSD50K (MAP) | | AudioSet (MAP) | |
|---|---|---|---|---|---|---|---|---|---|---|
| | ZS | LP | ZS | LP | ZS | LP | ZS | LP | ZS | LP |
| Supervise | N/A | 0.53 | N/A | 0.63 | N/A | 0.60 | N/A | 0.32 | N/A | 0.24 |
| YamNet | N/A | 0.85 | N/A | 0.78 | N/A | 0.63 | N/A | 0.50 | N/A | 0.27 |
| Wav2CLIP | 0.41 | 0.86 | 0.40 | 0.81 | 0.24 | 0.63 | 0.03 | 0.43 | 0.02 | 0.30 |
| AudioCLIP | 0.69 | **0.97** | 0.65 | **0.90** | 0.27 | 0.70 | 0.13 | 0.54 | 0.03 | 0.28 |
| CLAP | 0.77 | 0.96 | 0.73 | 0.88 | **0.30** | 0.72 | 0.14 | 0.59 | 0.06 | 0.32 |
| CoLLAT | **0.84** | **0.97** | **0.77** | 0.89 | 0.29 | **0.74** | **0.19** | **0.64** | **0.09** | **0.39** |

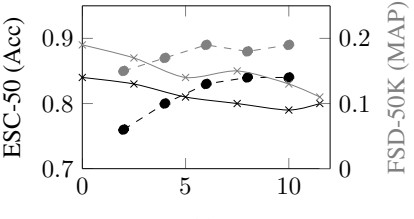

| Backbone Audio Encoder | ESC-50 (Acc) | FSD-50K (MAP) |
|---|---|---|
| ResNet [35] | 0.80 | 0.14 |
| CNN14 [17] | 0.82 | 0.16 |
| AST (Random Initialization) [10] | 0.82 | 0.18 |
| AST (Pretrained Initialization) [10] | **0.84** | **0.19** |

(a)                                                                  (b)

Figure 4: (a) Zero-shot audio classification performance of CoLLAT for ESC-50 and FSD-50K: **solid line** - when different intermediate layers are used to compute the token alignment loss (x axis denotes the half of the index of the layer used to compute the loss); and **dashed line** - when the number of cross-attention blocks in the token interaction module varies (x axis denotes the number of cross attention blocks). (b) Ablation study with different backbone audio networks using ZS audio classification. The pre-trained AST was pre-trained for audio classification using AudioSet.

For linear probe (LP) setting, we freeze our audio encoder as feature extractor and only train a 1-layer transformer decoder to predict the class labels.

Our results show that the Supervise baseline does not perform well across all datasets, which suggests that audio-text pre-training generally improves performance, particularly for tasks with limited labeled data (such as ESC-50). Among the other baselines, CoLLAT outperforms them mostly across all datasets. In particular, performance improvements from CoLLAT are significant for datasets that consist of complex audio clips with multiple sound clips such as AudioSet and FSD50K. For instance, CoLLAT outperforms the strongest baseline for the AudioSet dataset by 50% under zero-shot setting and 22% under linear probe setting. This observation highlights the superiority of CoLLAT in comprehending complex audio clips with multiple audio concepts.

**Ablation Study using Zero-shot Audio Classification.** In Fig. 4a, we provide ablations with different layer selections for token-level loss computation and the performance of CoLLAT with different number of cross-attention blocks. This ablation study guided to adopt the initial token-level layer in the text encoder to compute the token-level loss functions and 8 cross-attention blocks in the final architecture of the CoLLAT's token interaction module. Table 4b presents CoLLAT's performance with different audio encoder choices, which shows pre-trained AST as the most suitable backbone architecture. Additionally, we noticed that the AST backbone architecture converges faster compared

Table 3: (a) Ablation of loss functions using Zero-shot Audio Classification with clean and noisy audio samples; (b) Results for Cross-modal Retrieval, following the experimental setups in [35] and [36] for VGGSound and AudioCaps respectively. Higher is better for all the figures.

| | ESC-50 (Acc) | | FSD-50K (MAP) | |
|---|---|---|---|---|
| | clean | noisy | clean | noisy |
| CoLLAT | 0.84 | **0.59** | **0.19** | **0.17** |
| $(-)L_{cross-token}$ | 0.81 | 0.50 | 0.13 | 0.11 |
| $(-)L_{uni-token}$ | **0.85** | 0.53 | 0.18 | 0.13 |
| $(-)L_{cross-global}$ | 0.82 | 0.54 | 0.17 | 0.16 |

| | VGGSound (MRR) | | AudioCaps (MAP@10) | |
|---|---|---|---|---|
| | A→I | I→A | A→T | T→A |
| Wav2CLIP | 0.057 | 0.068 | 0.52 | 0.38 |
| AudioCLIP | 0.060 | 0.073 | 0.58 | 0.52 |
| CLAP | 0.063 | 0.074 | 0.57 | 0.47 |
| CoLLAT | **0.093** | **0.112** | **0.62** | **0.59** |

(a)                                                                  (b)

Table 4: Results for Audio Captioning using the AudioCaps dataset. The experimental setup freezes each model as a feature extractor and only train a 1-layer transformer decoder to predict text sequences. We follow the DCASE challenge setup [20] and report standard COCO caption evaluation metrics.

| | $BLEU_1$ | $BLEU_2$ | $BLEU_3$ | $BLEU_4$ | $CIDEr$ | $METEOR$ | $ROUGE$ | $SPICE$ |
|---|---|---|---|---|---|---|---|---|
| AudioCLIP | 50.2 | 34.9 | 22.1 | 14.4 | 14.4 | 28.7 | 36.1 | 8.7 |
| CLAP | 48.4 | 31.7 | 20.4 | 13.4 | 13.7 | 30.2 | 35.1 | 8.9 |
| CoLLAT | **55.1** | **37.2** | **24.3** | **15.1** | **16.3** | **43.0** | **39.6** | **11.3** |
| Human | 65.4 | 48.9 | 37.3 | 29.1 | 28.8 | 91.3 | 49.6 | 21.6 |

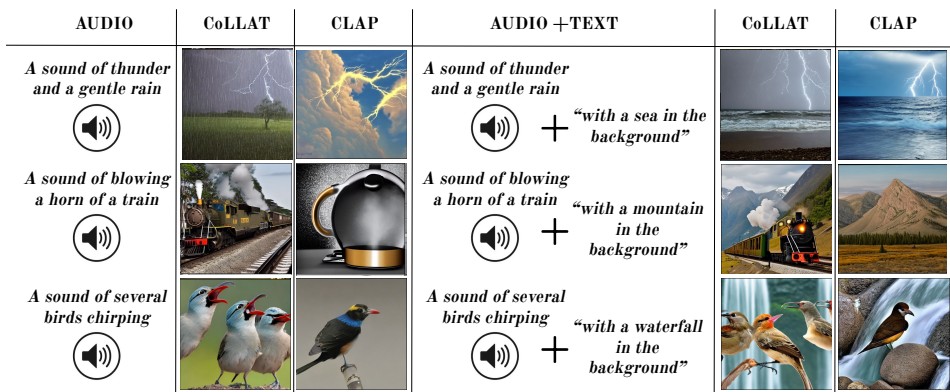

Figure 5: Audio-guided Image Generation Results

to the other backbones. This could be attributed to the stronger inductive biases present in the other encoders as compared to AST [10].

We conducted an ablation study in Table 3a to demonstrate the significance of the loss functions used in CoLLAT. The proposed token-level cross-modal loss function, $L_{cross-token}$, is shown to play a crucial role in enhancing audio-understanding performance. This is especially evident in FSD-50K, which consists of audio clips with multiple labels, where the performance for zero-shot audio classification is improved by $46.2\%$ in MAP. We also found that the robustness of CoLLAT substantially improves with the use of the unimodal-token level loss, $L_{uni-token}$, as it enhances the zero-shot classification performance for noisy audio samples that are generated by adding Gaussian noise with zero mean and 0.1 standard deviation. $L_{uni-token}$ improves the performance for such samples by $11.3\%$ and $30.8\%$ for ESC-50 and FSD-50K, respectively.

## 5.2 Cross-modal Retrieval

Since pretrained CLIP text encoder maps text to a joint embedding space of images and text, being able to map audio into the same CLIP embedding space (without fine-tuning the text encoder) enables cross-modal retrieving capabilities between audio, text and images. Table 3b shows the results collected for cross-modal retrieval task using the VGGSound and AudioCaps datasets. We adopt the experimental setups proposed in [35] for the audio-image and image-audio retrieval tasks and [36] for the audio-text and text-audio retrieval tasks. For a fair comparison, we train all the baselines with a frozen text encoder.

As can be seen, CoLLAT outperforms all the baselines by as much as $47.6\%$ and $51.4\%$ in Mean Reciprocal Rank (MRR) for the Audio-to-Image and Image-to-Audio retrieval tasks, and $13.4\%$ and $6.7\%$ in MAP for Audio-to-Text retrieval, and Text-to-Audio retrieval, respectively. We observed that the performance improvements are particularly significant due to the ability of our approach to understand fine-grained concepts to differentiate hard image-text pairs in the VGGSound.

## 5.3 Audio Captioning

We report the results for Audio Captioning using the AudioCaps dataset in Table. 4. Following [35], we only trained a 1-layer transformer on top of the CoLLAT's output, while keeping CoLLAT's parameters fixed to collect the results for audio captioning. We adopt the same experimental setting for the baselines too. As can be seen, CoLLAT outperforms CLAP and AudioCLIP by 9.7%, 13.2%,

42.4%, 9.7% and 26.9% in $BLEU_1$, $CIDEr$, $METEOR$, $ROUGE$ and $SPICE$ respectively. These results further show the ability of CoLLAT fine-grained details in the sound clip as it is needed to generate a holistic caption for a given audio clip.

### 5.4 Audio-guided Image Generation

Given CoLLAT's ability to produce corresponding text token-level embeddings for a given audio clip, it can be naively used to provide guidance for any downstream application built on top of text token embeddings from pretrained text encoders. We adopt text-guided image generation as one such example. Most existing text-guided image generation models adopt token-level embeddings from a pretrained text encoder to guide image generation using a given text prompt. Nevertheless, most existing audio-to-text grounding techniques cannot be naively extended to introduce audio guidance for such applications, as they are unable to generate token-level text embeddings for a given audio prompt.

To evaluate this task, we adopt a pretrained text-guided image generation model following [29], which requires text guidance from a pretrained CLIP text encoder. As a baseline, we adopt the same architecture as CoLLAT, but trained only using $L_{cross-global}$. We denote this baseline as CLAP in Figure 5. As shown in Figure 5, CoLLAT is capable of generating images that cover all the concepts present in the given audio clip. This observation further confirms the potential of our approach to comprehend complex audio clips with multiple concepts.

## 6 Broader Impact

Our work has a significant impact as it addresses the limitations of conventional audio understanding models by enhancing their ability to predict finer-grained classes within complex audio clips. We demonstrated that CoLLAT achieves state-of-the-art performance in audio understanding, as validated by its success in various downstream applications. Furthermore, it unlocks the potential for audio guidance in applications built upon pretrained language models. Thus, our work opens up possibilities for improved audio-to-text grounding, advancing multimodal applications. Additionally, considering the capability of CoLLAT to represent audio clips using either text or image, it holds promise in enhancing accessibility and promoting inclusivity for individuals with diverse disabilities.

It is important to note that the model developed in this research does not possess the ability to uniquely identify individuals in audio recordings for tasks such as speaker recognition and speaker identification. CoLLAT does not possess the ability to comprehend information conveyed through speech in audio clips. Given the privacy concerns associated with the generation of human faces using audio-guided image generation models, it is advised to exercise caution and refrain from utilizing them for such purposes. It is crucial to prioritize privacy and ethical considerations when applying these technologies to avoid any potential misuse or infringement on individuals' rights.

## 7 Conclusion

This work proposes CoLLAT, a novel framework for learning an audio understanding model using natural language supervision from a locked pretrained language model. The proposed framework effectively exploits the implicit knowledge in large language models instead of just relying on their final global embeddings. We introduce a novel pretraining objective for CoLLAT that enforces token-level alignment between audio and text modalities to yield fine-grained audio understanding. Our results demonstrate that CoLLAT achieves state-of-the-art performance for downstream applications such as audio classification, cross-modal retrieval and audio captioning. Since CoLLAT keeps the pre-trained language model locked during training, our experiments also show that it unlocks new applications such as audio-guided image generation.

Since CoLLAT can be considered a modality-agnostic framework for learning knowledge understanding models for new modalities with natural language supervision, extending CoLLAT to new modalities (video) is an interesting future direction to explore. Another notable limitation of CoLLAT is its inability to understand speech signals in the audio clips. Therefore, it could be worthwhile to explore how to introduce speech signals to CoLLAT while preserving its existing capabilities.

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
