# OpenReview forum: "CoLLAT: On Adding Fine-grained Audio Understanding to Language Models using Token-Level Locked-Language Tuning"
_NeurIPS.cc/2023/Conference — NeurIPS 2023 poster_

### Official Review · Reviewer_GZR1 · 2023-07-02

**Soundness:** 3 good
**Presentation:** 3 good
**Contribution:** 3 good
**Rating:** 6
**Confidence:** 4

**Summary:**

The authors propose a novel ways to train audio-text model by adding a token interaction module followed by the same frozen text encoder from the text modality after audio encoder to learn audio representation. And they show that proposed approach yield state-of-the-art results in couple audio tagging and classification tasks, cross-modal retrieval, and qualitative analysis on audio guided image generation.

**Strengths:**

- The proposed approach of appending token interaction module followed by text encoder is a very interesting and novel idea to leverage language model.
- Downstream evaluation are provided with both zero-shot and linear probe settings, this provides more thorough understanding of the behavior for proposed systems.

**Weaknesses:**

- Proposed architecture is really interesting, however, the training with AudioSet labels in constructing prompts as described in section 4.1 line 248-250 seem arbitrary. The claim that "such templates have been shown to be more effective than simple concatenation" needs more in depth evidence or reference. Also given the works in increasing audio captioning data and applied to audio-text model pre-training (e.g. [1], [2]), this work can benefit from pre-training utilizing these audio and natural language descriptions besides pre-training with AudioSet with arbitrary templating on labels.
- On the downstream tasks, other than audio tagging/classification tasks, language-based audio retrieval is another highly relevant task which provide different perspective to the audio-text pre-training, consider adding those audio-text benchmarks [2] to the evaluation to expand the coverage in this work.

[1] Elizalde, Benjamin, et al. "Clap learning audio concepts from natural language supervision." ICASSP 2023-2023 IEEE International Conference on Acoustics, Speech and Signal Processing (ICASSP). IEEE, 2023.

[2] Wu, Yusong, et al. "Large-scale contrastive language-audio pretraining with feature fusion and keyword-to-caption augmentation." ICASSP 2023-2023 IEEE International Conference on Acoustics, Speech and Signal Processing (ICASSP). IEEE, 2023.

**Questions:**

- For the audio and text encoder selections, have you tried other options, for example ResNet/CNN based for audio and other text encoders such as BERT or RoBERTa? For the other systems to compare as shown in Table 2, they all utilize different encoder architectures, such ablation study can help improve and disentangle whether the improvement comes from proposed architecture versus encoder choices.

**Limitations:**

- This work is slightly limited in then evaluation with only audio tagging/classification tasks, it can benefit from adding more audio-text tasks involving longer form natural languages to provide more holistic views.

---

> ### Author Rebuttal · Authors · 2023-08-09
>
> We appreciate your valuable comments and feedback. Please refer to the attached PDF in the global response for supplementary results.
>
> __1. About the selected template__ Following the zero-shot performance of the strongest baselines in our experiments [1], we employ the template in Section 4.1 rather than simple concatenation. Please refer to Table 3 in [1] for a comparison of performance using different templates. Adopting this approach enables us to demonstrate that the performance improvement achieved by CoLLAT over CLAP is not attributed to the template selection. We will clearly cite [1] in the relevant section of the paper.
>
> We made the decision to not utilize natural language descriptions or a keyword-to-caption augmentation [2] in favor of simple templates for the following reasons:
> * The cross-modal token-level loss function in CoLLAT aims to explicitly map each token in the given text prompt with its corresponding counterpart in the audio tokens (i.e., the patches in the Mel-Spectrogram of the audio). It may not be feasible to find such a mapping for certain tokens in natural language descriptions (e.g., stop words, adjectives) with the audio tokens. Consequently, having complex text prompts could adversely impact the training of CoLLAT, particularly given the greedy algorithm we utilized to compute $g$ for mapping audio tokens with text tokens.
> *  CoLLAT maintains the text encoder frozen during training, as it was pre-trained using text labels from the LAION dataset rather than natural language descriptions. Thus, attempting to produce text embeddings for natural language descriptions using the CLIP text encoder to train CoLLAT could introduce a data shift problem, which could also negatively impact the training of CoLLAT.
> * The scale of the selected training dataset, AudioSet, which consists of approximately 2 million audio and text label pairs, is significantly larger compared to datasets with audio and text captions. In fact, it is more than three times larger than the largest publicly available dataset of this kind, LAION-630K.
>
> But we acknowledge that these points should be validated with sufficient evidence; however, due to time constraints, we leave this validation as future work.
>
> [1] Elizalde, Benjamin, et al. "Clap learning audio concepts from natural language supervision." ICASSP 2023-2023 IEEE International Conference on Acoustics, Speech and Signal Processing (ICASSP). IEEE, 2023.
>
> [2] Wu, Yusong, et al. "Large-scale contrastive language-audio pretraining with feature fusion and keyword-to-caption augmentation." ICASSP 2023-2023 IEEE International Conference on Acoustics, Speech and Signal Processing (ICASSP). IEEE, 2023.
>
> __2. Results for several additional tasks__ Please find the results for text-to-audio retrieval, audio-to-text retrieval, and audio captioning using the AudioCaps dataset in Figure 1.b and Table 1 in the supplementary PDF in the global response. In the latest results, CoLLAT outperforms the baselines (by approximately 20% in SPICE for audio captioning, 13.4% in MAP for audio-to-text retrieval, and 6.7% in MAP for text-to-audio retrieval), demonstrating a similar performance trend as the previously reported results in the main paper.
>
> __3. Ablations with different encoder choices__  We present the results with different choices of audio encoders in Figure 1.b of the supplementary PDF in the rebuttal. Our observations show a slight performance improvement with the AST backbone when using a pretrained initialization. Additionally, we noticed that the AST backbone architecture converges faster compared to the other backbones. This could be attributed to the stronger inductive biases present in the other encoders as compared to AST [3].
>
> Due to time constraints, we leave the performance evaluation with text encoders for future work. We anticipate the results to be consistent across different transformer-based architectures such as BERT, ROBERTA, and CLIP, provided they are pre-trained on the same contrastive objective as CLIP. However, we acknowledge that additional experiments are necessary to evaluate performance across different text encoders trained on various pre-training objectives (e.g., masked language modeling). Considering that most downstream applications, such as text-to-image generation, rely on the text conditioning from CLIP, we have chosen it as our main text encoder. This decision aligns with one of the primary motivations of this work, which is to cost-effectively introduce audio guidance to downstream applications that are built on text guidance from pre-trained text encoders. Please also note that the majority of baselines in our study adopt the same text encoder.
>
> [3] Gong, Yuan, Yu-An Chung, and James Glass. "Ast: Audio spectrogram transformer." arXiv preprint arXiv:2104.01778 (2021).

---

> > ### Comment · Reviewer_GZR1 · 2023-08-20
> >
> > Thank the authors for your responses and extra experiments in audio captioning and text to audio retrieval. It would be great to include these results in the final paper if possible to provide more thorough view of your work. I maintain my rating and advocate for accepting of this work.

---

> > > ### Author Response · Authors · 2023-08-21
> > >
> > > We thank you for your effort in reviewing the rebuttal. We will incorporate all of your comments into the final version of the paper.

---

### Official Review · Reviewer_Ujf9 · 2023-07-05

**Soundness:** 3 good
**Presentation:** 3 good
**Contribution:** 3 good
**Rating:** 6
**Confidence:** 3

**Summary:**

This paper presents a novel approach to train audio embeddings grounded in text embeddings for audio classification. They propose a trainable audio embedding layer while keeping the pre-trained text embedding module frozen. Apart from the contrastive loss term that maximises similarity between corresponding audio and text embeddings, authors propose a loss term that exploits audio-text alignment at token level. They also propose a loss term that minimizes the distance between embedding obtained for clean vs noisy audio sample. The authors show that their model outperforms other SOTA models for audio classification and audio retrieval.

**Strengths:**

One of the key strengths of the paper is that they freeze the pre-trained text embedding module and pull the audio embedding closer to the text embeddings obtained from the frozen module. This enables their audio embedding module to convert audio waveforms to text embeddings which can be recognized by other models using the pre-trained text module. This enabled them to showcase the efficacy of their method for audio guided image generation using pre-trained diffusion models. The proposed loss terms are well motivated. They do extensive evaluations comparing their model with other baselines and show it beats them for the two tasks of audio classification and cross-modal retrieval. The ablation studies show case the benefit of each of the proposed modules. The audio guided image generation further shows the benefit of cross-modal token level alignment loss.

**Weaknesses:**

1. Not enough details are provided to reproduce the experiments e.g. detail of their AST backbone for audio encoder, architecture details of token interaction module.
2. Functionality of the permutation module “g” in cross-modal token level alignment loss is not very clear (equation 1). Is the permutation “g” different for different training examples? Is it different for the same example across different epochs? What is the permutation used in inference time? Or do the authors not permute the token embedding generated from audio during inference?
3. For cross-modal token-level alignment loss authors do a permutation of the token level embeddings obtained from audio before matching them with text tokens. For a length N sequence, there would be N! ways the sequence can be arranged. It would seem that would lead to a significant increase in training time trying to find the permutation that leads to the lowest loss especially when done for all the examples in a batch for all epochs.


**Questions:**

1. For token interaction module, was the token sequence of length N provided as input (query) to first layer initialized randomly?
2. Learning rate of 0.1 for Adam seems high. Clap uses 1e-3 while AST used 5e-5. Did the authors see unstable training with such high learning rate?
3. Did authors try with loss terms minimizing the distance between token embedding from audio and actual text embeddings obtained from intermediate layers of the pre-trained text encoder?
4. Why did authors choose to use learned position embeddings? Previous research has shown that using fixed sinusoidal position embedding o rotary position embeddings can also work well with transformer architectures. Did authors try using fixed position embeddings?

**Limitations:**

Authors have addressed the concerns about identifying personal information from audio recording. they have also considered the possibility that one can potentially use audio guided image generation to generate human faces which can lead to privacy and ethical issues. In my opinion they have adequately addressed the ethical concerns.

---

> ### Author Rebuttal · Authors · 2023-08-09
>
> We appreciate your valuable comments and feedback. Please refer to the attached PDF in the global response for supplementary results.
>
> __1. Initialization of the input to the token interaction module__ All elements in the input token embedding matrix were randomly initialized, following a normal distribution with a mean of zero and a standard deviation of 1. We will provide a clearer explanation of this point in Line 195 of the main paper.
>
> __2. Learning rate__ We did not observe significant instabilities with the selected learning rate of 0.1, combined with the other optimal hyperparameters. During the initial iterations with a small learning rate, it was observed that the absolute values of parameter updates could be relatively small. This behavior was more noticeable when utilizing a later layer of the text encoder to compute the token-level loss functions. This behavior can be attributed to two factors: (1) the strong dependence of the optimal learning rate on the objective functions and the weights assigned to them (lambda values), rather than the architecture; and (2) the better pre-trained initialization of most components in CoLLAT.
>
> Please also note that the chosen learning rate serves as an upper bound. The learning rate is adaptively adjusted for each parameter when using the Adam optimizer. However, we observed instances of instabilities and encountered challenges related to reaching trivial solutions (e.g., averaged embedding as the token embeddings) when training CoLLAT solely with the cross-modal token alignment loss without applying $g$.
>
> __3. Results when different intermediate layers are used to compute the token-level alignment loss__ We present the results for this study in Figure 2.a. As can be see, we have identified that the first layer is the optimal layer for performing token-level alignment. This finding may be attributed to the following reasons:
> * As CoLLAT utilizes a frozen text encoder in both the text and audio encoding pipelines, performing cross-modal token-level alignment at an earlier layer of the frozen text encoder inherently aligns the token embeddings in subsequent layers as well.
> * The token embeddings from the intermediate layers in CLIP are more contextualized, resulting in less distinct embeddings for different tokens due to the information-sharing with their contextual surroundings. CoLLAT effectively leverage this by employing the first layer of CLIP (i.e., static embeddings of CLIP) to compute the cross-modal token-level loss function. This approach enhances the likelihood of achieving optimal reordering with our greedy reordering method to compute $g$, as the token embeddings show greater differences at the earliest layers.
>
> __4. Regarding positional embeddings__ The decision to adopt learnable positional embedding in CoLLAT was primarily motivated by AST [1], as we employed the pre-trained checkpoint of AST during our experiments. In our audio encoder, the patches of the audio Mel spectrogram are treated as tokens. Given the existence of long-distance and somewhat arbitrary spatial dependencies among such tokens, the adoption of learnable positional embedding is widely accepted in similar domains [1,2]. For our text encoder, we follow the same positional embedding technique as CLIP, as we keep our text encoder fixed during training.
>
> [1] Gong, Yuan, Yu-An Chung, and James Glass. "Ast: Audio spectrogram transformer." arXiv preprint arXiv:2104.01778 (2021).
>
> [2] Dosovitskiy, Alexey, et al. "An image is worth 16x16 words: Transformers for image recognition at scale." arXiv preprint arXiv:2010.11929 (2020).
>
> __5 More details about the architecture__ We adopt the same architecture as AST to define our audio encoder. The cross-attention block in the token-interaction module is architecturally similar to the cross-attention block proposed in [3]. Our CLIP text encoder is taken from the CLIP encoder in the Stable Diffusion V2 model. To ensure the paper is self-contained, we will include these specific details in the supplementary material of the final version. In the supplementary material of the rebuttal, we present the ablations (see Fig. 2b) that we conducted to determine the optimal number of cross-attention blocks in the token-interaction module. Following this study, we set the number of cross-attention blocks to 8. Additionally, in Figure 1.b of the supplementary material, we provide the results obtained using different audio encoders, which led us to select AST as our backbone architecture.
>
> [3] Chen-Wei Xie, Jianmin Wu, Yun Zheng, Pan Pan, and Xian-Sheng Hua. Token embeddings alignment for cross-modal retrieval. In Proc. of ACM-MM, 2022.
>
> __6 More details about the functionality of $g$__ The introduction of $g$ in the cross-modal token alignment loss allows CoLLAT to be agnostic to the order of classes within the text. As a result, CoLLAT's objective solely focuses on generating all the textual concepts from a given text prompt using the associated audio clip, without penalizing incorrect ordering. Since there are multiple ways to concatenate the text labels of an audio clip to form a text prompt, CoLLAT's token interaction module cannot predict the corresponding text tokens in the correct order. This is because the module does not have access to information regarding the specific order used to create the text prompt. Providing such information to the token interaction module can lead to inference-related problems such as which order should be used during inference. The permutation $g$ is different for different training examples and even for the same example across different epochs, especially during the early stages of training. This permutation is necessary only for computing the cross-model token alignment loss function and is not required for inference (obtaining the audio embedding). We agree with you regarding the complexity of $g$, which is N! for each instance, and it acts as the bottleneck in the training loop of CoLLAT.

---

> > ### Comment · Reviewer_Ujf9 · 2023-08-17
> >
> > I thank the authors for their revision. The authors have adequately answered my questions and concerns. I also thank them for providing additional experiments in the supplementary material showing the results on AudioCaps and providing more ablation studies. I think the methodology is well motivated and it could find applications in areas where fine-tuning a big text encoder like CLIP would be bottleneck. Training time is however still a concern due to the permutation operation. I raise my review score from 5 to 6.

---

> > > ### Author Response · Authors · 2023-08-20
> > >
> > > We thank you for your effort in reviewing the rebuttal. We will incorporate all of your comments into the final version of the paper.

---

### Official Review · Reviewer_NoKh · 2023-07-06

**Soundness:** 3 good
**Presentation:** 3 good
**Contribution:** 3 good
**Rating:** 7
**Confidence:** 3

**Summary:**

This paper introduces a method to train an audio encoder to produce audio embeddings that match text embeddings. Contrary to other methods, the authors suggest to keep the text encoder fixed so as to enable its use in other contexts. More specifically, a Transformer audio encoder produces audio embeddings. A second model produces audio-to-text embeddings with the same dimension as the reference text, which are re-ordered and matched with the text embeddings. This module allows the original audio embeddings to have a fine-grained correspondance with the text. Finally the outputs of the frozen text encoder applied to the reference text embeddings and the embeddings derived from the audio-to-text model are matched with a contrastive loss. The experiments show good results for audio classification tasks as well as cross-modal retrieval and audio-guided image generation.

**Strengths:**

  - The paper is clear and convincing. The introduction and related work section are especially well written and clear, containing a good presentation of the challenges and contributions
  - The motivation to keep the pre-trained text encoder untouched is clear and interesting. The results are a good proof of the interest of this approach. The examples illustrate well the effectiveness of the method.
  - The idea to produce fine-grained correspondance between the audio embeddings and the text is reasonable and pretty good.
  - The experiments look sound and are convincing

**Weaknesses:**

  - The presentation and notation could be more explicit at some places, for example:
    - Intro, 2nd paragraph: "as shown in fig 1b" : not clear what to look at in 1b. Maybe explain which one is the fine-tuned CLIP?
    - it looks like there is a formatting issue in 3.1
    - in 3.3, the notation requires more explanation. It looks like the two S are transpose of each other.
    - Eqn. 2 requires more explanation. what is the (i, i) subscript? softmax along which dimension of S?
  - It would be nice to see some examples of classes/text in different datasets (maybe in supplementary material) to illustrate the complexity of datasets
  - Some parts could benefit from more explanations, for example:
    - How is the reordering $g$ computed?
    - How are the initial input defined in the denoising pipeline?
  - It would be interesting to have a deeper study and analysis of the results in Fig. 2. While reading, I expected to see the corresponding experiments and analysis in the experiment section.


Minor:
  - l.107: "correspndence."
  - formatting issue in the text in Sec. 3.1 making the section hard to read and understand
  - Missing reference for AudioSet
  - Evaluation: missing "(3)" for the third task
  - Sec. "4.1 Datasets" does not only deal with datasets. There is not Sec. 4.2

**Questions:**

3.1:  how are the initial input defined in the denoising pipeline?

3.2: greedy reordering: how much better would the method be if the optimal reordering was found? could the reordering be learned by the token interaction module, or why isn't it? How the order of the classes in the text and in the audio relate in the training set?

4.1: In the template, does the ordering of the classes matter? (e.g. if the classes appear one after the other in the audio)

Sec. 5.1: Zero Shot: is the possible set of labels in the text encoder also matching the template presented in 4.1?



**Limitations:**

yes

---

> ### Author Rebuttal · Authors · 2023-08-08
>
> We appreciate your valuable comments and feedback. Please refer to the attached PDF in the global response for supplementary results.
>
> __1. Unclarity about the fine-tuned CLIP models in Fig. 1b__ In Fig. 1b, CLAP, AudioCLIP, and Cons-CLAP models fine-tune the pre-trained CLIP text encoder. In contrast to CLAP and AudioCLIP, Cons-CLAP incorporates a constraint to alleviate catastrophic forgetting of CLIP's existing knowledge during training. This constraint penalizes the model if it significantly forgets existing knowledge. In the final version, we will clearly specify and highlight these points.
>
> __2. Formatting issue in Section 3__ Thank you so much for pointing out these issues. In Sec. 3.1, the notations should be corrected as bellow. We will correct these issues in the final version.
>
> ${x^a_i}{i=1}^M \rightarrow$ {$x^a_i$}$_{i=1}^M$
>
> ${x^t_j}{j=1}^N \rightarrow$ {$x^t_j$}$_{j=1}^N$
>
> ${h^a_i}{i=1}^M \rightarrow$ {$h^a_i$}$_{i=1}^M$
>
> ${h^t_j}{j=1}^N \rightarrow$ {$h^t_j$}$_{j=1}^N$
>
> __3. Explanation on $S$ in Section 3.3__ In Section 3.3, two $S$ matrices are $|B|\times |B|$ matrices that are transposes of each other, as you mentioned. Here, $|B|$ refers to the batch size. The $(i,i)$ subscript of the $S$ matrices represents the elements along the main diagonal. The elements along the main diagonal in $S$ represent the dot product similarity of the embeddings between corresponding audio and text pairs. We will make these facts clear in the final version of the paper.
>
> __4. Few examples from different datasets__ We will include a few examples from each dataset and detailed statistics related to complexity (such as annotator agreement) in the final supplementary material of the paper.
>
> __5. About how $g$ is computed__ $g$ aims to find the permutation of the token-level embeddings in {$h^{a-t}_j$}$_1^N$ that minimizes the L1-norm distance between the permuted audio token embeddings of an audio clip and the corresponding text token embeddings. $g$ is computed greedily given the time complexity of finding the optimal solution for $g$. Our greedy algorithm operates as follows: Given the token embeddings of {$h^{a-t}_j$}$_1^N$ and {$h^{t}_j$}$_1^N$, we initiate the process by starting from the right-most token embedding in {$h^{t}_j$}$_1^N$. This token is then paired with the closest token embedding in {$h^{a-t}_j$}$_1^N$. Next, we proceed to the second token embedding from the right in {$h^{t}_j$}$_1^N$, excluding the already mapped token embedding in {$h^{a-t}_j$}$_1^N$ from the possible set to be paired with the selected text token embedding. We iterate this process until we obtain the greedy one-to-one mapping between all the tokens in {$h^{a-t}_j$}$_1^N$ and {$h^{t}_j$}$_1^N$. We observed that this greedy approach could produce optimal re-ordering unless there are very similar classes in the same audio clip. Consequently, we do not anticipate a significant variation in the performance if the optimal reordering can be achieved, which we leave as future work given the time complexity of training the model with the optimal reordering.
>
> __6. Initialization of the input to the denoising pipeline in the token interaction module__ All the elements in the input matrix were randomly initialized by following a normal distribution with a mean of zero and a standard deviation of 1.
>
> __7. Impact of the order of classes in text prompts__ CoLLAT is agnostic to the order of the classes presented in the text. This is the primary motivation behind the introduction of $g$, which reorders the token embeddings in {$h^{a-t}_j$}$_1^N$ prior to computing the token alignment loss. With the aid of $g$, CoLLAT's objective is solely focused on producing all the textual concepts from a given text prompt using the corresponding audio clip, but incorrect ordering is not penalized. Consequently, the ordering of class in template in 4.1 does not matter.
>
> For a given audio clip with multiple labels, there are multiple ways to concatenate these labels together and generate the text prompt using the template described in Section 4.1. However, since we do not provide any information about the specific order utilized to create the text prompt to our token interaction module (providing such information to the token interaction module causes problems during inference), it is not possible for the module to learn the reordering process for a given (audio, text) pair by itself. This is the main motivation to make CoLLAT agnostic to the order of the classes in the text prompt.
>
> __8. Impact of the oder of sound events in audio clips__ Due to cross-attention mechanism in CoLLAT's token interaction module, CoLLAT remains agnostic to the order of classes present in the audio. Even in cases where classes overlap within the same time period, CoLLAT can accurately map them to their corresponding text counterparts if their frequencies differ. This capability is possible because CoLLAT's audio encoder treats each patch in the Mel-spectrogram of an audio clip as a distinct audio token.
>
> __9. About Zero Shot setting__ Yes, the possible set of labels is also processed using the same template as described in Section 4.1 (_This is a sound of [class label]_) before being inputted into the text encoder to obtain the corresponding text embeddings.

---

### Official Review · Reviewer_swwz · 2023-07-06

**Soundness:** 3 good
**Presentation:** 4 excellent
**Contribution:** 3 good
**Rating:** 6
**Confidence:** 4

**Summary:**

The paper introduces CoLLAT, an audio-language framework that makes use of a pre-trained language model. The framework is trained with a contrastive objective which encourages learning of fine-grained audio-text grounding. The paper presents very strong results for diverse downstream tasks, such as zero-shot audio classification, audio-image retrieval, and audio-guided image generation.

**Strengths:**

The proposed framework leverages a locked pre-trained language model (CLIP text encoder) as audio and text encoder. It learns to map the audio input to input tokens for the frozen text encoder, which retains the strong capabilities of the language model. The model is trained to encourage very fine-grained understanding of audio. It achieves very convincing results on various downstream tasks.

**Weaknesses:**

Given that this paper claims to yield fine-grained audio-to-text grounding, I believe that it would be good to show results for audio-text retrieval on CLOTHO, AudioCaps, and SoundDescs [A,B].

One of the main model contributions, the token interaction module, should be explained better and in more detail. For instance, it is not mentioned how many blocks are used (l.192: “multiple cross-attention-based blocks”). In addition, it would be good to add a detailed explanation on the denoising pipeline in this module. Furthermore, different components of this block should be ablated in the model ablation study.
Similarly the paper does currently not contain a model ablation for 1) the use of the same encoder for audio and text instead of using a different audio model, 2) the effect of using pre-trained encoders. These should be added to validate the architectural choices.

Overall, I think the paper is interesting, and I would be happy to consider raising my score if the missing evaluations and ablations are added and convincing.



[A]: Oncescu et al.: Audio Retrieval with Natural Language Queries, INTERSPEECH 2021

[B] Koepke et al.: Audio Retrieval with Natural Language Queries: A benchmark study, Transactions on Multimedia 2022

**Questions:**

* Is CLAP shown in Figure 5 referring to [6]? If not, different naming should be used.

* I believe that some relevant prior works should be referenced, e.g. [C,D,E].

* What is the computational cost to train the framework and what kind of hardware was used for training?

[C]: Wu et al: LARGE-SCALE CONTRASTIVE LANGUAGE-AUDIO PRETRAINING WITH FEATURE FUSION AND KEYWORD-TO-CAPTION AUGMENTATION, ICASSP 2022

​​[D]: Deshmukh et al.: Audio retrieval with wavtext5k and CLAP training, 2022

[E]: Wu et a.: Large-scale Contrastive Language-Audio Pretraining with Feature Fusion and Keyword-to-Caption Augmentation, ICASSP 2023



* Typos:
l.62: lost -> lose

l.136 alignemnt -> alignment

l.181: space missing after “CLIP.”

l.232: We adopt AudioSet dataset -> We adopt the AudioSet dataset

Figure 4 is a table and should be labelled accordingly.

What does t in $128x100 t$ in l.240 refer to?

**Limitations:**

Yes.

---

> ### Author Rebuttal · Authors · 2023-08-08
>
> We appreciate your valuable comments and feedback. Please refer to the attached PDF in the global response for supplementary results.
>
> __1. Results for several additional tasks.__ Please find the results for text-to-audio retrieval, audio-to-text retrieval, and audio captioning using the AudioCaps dataset in Figure 1.b and Table 1 in the supplementary PDF in the global response. In the latest results, CoLLAT outperforms the baselines (by approximately 20% in SPICE for audio captioning, 13.4% in MAP for audio-to-text retrieval, and 6.7% in MAP for text-to-audio retrieval), demonstrating a similar performance trend as the previously reported results in the main paper. Due to time constraints, we will consider collecting results with CLOTHO and SoundDescs as future work.
>
> Please take note that the experimental setup for the zero-shot column in the audio classification results table in the main paper is similar to the audio-to-text retrieval, as it solely depends on the similarity of embeddings between text and audio to identify corresponding pairs, without utilizing any downstream classifier as in the LP setting. Therefore, the zero-shot audio classification results can also be considered as supplementary results for audio-to-text retrieval.
>
> __2. More ablations.__ In the supplementary PDF of the rebuttal, we provide additional ablations with different audio encoder choices (refer to Fig. 1.b), varying numbers of cross-attention blocks (refer to Fig. 2.b), and distinct layer selections for token-level loss computation (refer to Fig. 2.a). These ablations were conducted to guide different design decisions in CoLLAT. In our final architecture, we have incorporated 8 cross-attention blocks within the token-interaction module. We will clearly mention all these hyperparameters in the final version.
>
> __2.1. Impact of using the same encoder for audio and text instead of using a different audio model__ CoLLAT's architecture adopts the same frozen text encoder in the audio encoding pipeline (refer to Fig. 3 in the main paper). Considering the nature of the token-level loss functions, if we employ completely different models (without any parameter sharing) for audio and text encoding, it needs to train an audio encoder that matches the scale of CLIP (with over 1B parameters) for audio encoding. This learning process is resource-intensive, especially given the added complexity of $g$ in the cross-modal token-level loss function. To tackle this challenge, CoLLAT proposes a cost-effective alternative by efficiently sharing the parameters of a pre-trained text encoder within its audio encoding pipeline. With this architecture, our audio encoder can effectively utilize all the language understanding capabilities of the pre-trained text encoder when processing audio. This may explain why CLIP achieves significantly superior results in tasks like audio captioning compared to the baseline, which employs different models for audio and text encoding.
>
> __2.2. Impact of using pre-trained encoders__ Adopting pre-trained encoding heads to initialize audio encoders has been shown to be useful in previous similar works (e.g., AudioCLIP). We observe a similar trend for the ESC-50 and FSD-50K datasets with CoLLAT, which is reported in Fig. 1.b of the supplementary PDF. We also noticed that pre-trained initialization improves the convergence speed.
>
> __3. Differences between CLAP and the model used in Figure 5__ To effectively utilize the denoising diffusion pipeline for image generation as a readily available solution without re-training, two key requirements need to be met: (1) the text encoder must remain frozen during the audio encoder training, and (2) the audio encoder should accurately predict token embeddings of the corresponding text prompt for a given audio clip. This is mainly because the selected denoising diffusion model accepts token-level embeddings of either the text prompt or audio clip to guide the image generation process. To fulfill these requirements, we adopt CoLLAT's architecture and CLAP's training objective to train the baseline model denoted as CLAP in Fig. 5. For further details, please refer to lines 297 to 305 in the main paper. In the final version, we will ensure that different naming is used for this baseline model.
>
> __4. About the missing citations__ Thank you for pointing out these missing citations. It seems that the same work was mistakenly pasted for both [C] and [E]. We would greatly appreciate it if you could let us know if you have found two different works that should be cited for [C] and [E]. We will ensure that all these works are properly cited in the final version.
>
> __5. Computational cost and hardware__ We utilize a GPU cluster comprised of 8 V100 GPU cards for training CoLLAT. With this hardware configuration, it takes approximately 320 GPU hours to train the model on AudioSet using the optimal hyperparameter setting. We will include these details in the main paper.
>
> __6. Formatting issues__ Thank you for pointing the formatting issues and typos. We will correct all of these issues in the final version.
>
> __7. Meaning of $t$ in $128x100t$__ Here, t is the time duration of the audio clip in seconds. We will define this notation clearly in the final version.

---

> > ### Comment · Reviewer_swwz · 2023-08-19
> >
> > I would like to thank the authors for clarifying my doubts and for providing additional convincing experiments, in particular the additional ablations and tasks. Therefore, I vote to accept the paper.
> >
> > However, I urge the authors to incorporate all the additional experiments and clarifications in a final version of the paper. Additionally, the state-of-the-art performances should be included in the tables. In particular, Table 1a in the additionally provided pdf should include the sota results as an upper bound along with reporting not only R@10 but also other commonly reported metrics (e.g. R@1,.. etc).

---

> > > ### Author Response · Authors · 2023-08-20
> > >
> > > We thank you for your effort in reviewing the rebuttal. We will incorporate all of your comments into the final version of the paper.

---

### Author Rebuttal · Authors · 2023-08-09

We thank all the reviewers for their valuable comments and feedback. Kindly find the attached PDF in this global response for additional results that support our rebuttal.

---

### Decision · Program_Chairs · 2023-09-21

**Decision:**

Accept (poster)

**Comment:**

All reviewers have agreed that this paper has sufficient novelty and is worthy of acceptance.

This paper proposes an audio-language framework using pre-trained models and encourages fine-grained audio-text alignment. The authors conducted diverse experiments to justify the effectiveness. After all concerns were addressed by the authors (e.g. adding more ablation and more tasks for better justification), all reviewers voted to accept this paper. The authors also promised the reviewers to include more experiments in the camera-ready revision to make it more comprehensive.